# Design Method of Freeform Off-Axis Multi-Mirror Optical Systems

**Xinyu Liu and Jun Zhu \***

State Key Laboratory of Precision Measurement Technology and Instruments, Department of Precision Instrument, Tsinghua University, Beijing 100084, China; xy-liu17@mails.tsinghua.edu.cn

\* Correspondence: j_zhu@tsinghua.edu.cn

**Abstract:** A data point calculation method that does not require the use of Fermat′s principle and a simple and general design method of starting points of freeform off-axis multi-mirror optical systems are proposed in this paper, which aim to promote the realization of high-performance reflective systems containing freeform surfaces. Taking a planar system and the required parameters as the input, a good starting point for a freeform off-axis multi-mirror system can be automatically obtained using the proposed method. The design of a freeform off-axis five-mirror system with a low F-number is taken as an example to show the effectiveness of the proposed method. The method can also be used for the design of freeform reflective systems with other numbers of mirrors.

**Keywords:** imaging system; freeform; multi-mirror



## 1. Introduction

Freeform surfaces can be defined as surfaces without an axis of rotational symmetry [1], which have stronger aberration correction abilities than rotationally symmetric surfaces. Freeform optics is an emerging technology that has revolutionized imaging and non-imaging optics [2,3], which has enabled high-performance imaging systems and novel non-imaging systems [4–6]. In imaging optics, freeform surfaces have been used successfully in telescopes [7–10], spectrometers [11–14] and head-mounted displays [15–17]. Freeform surfaces have broad application prospects in both refractive and reflective systems. Among them, freeform off-axis reflective optical systems have interested many optical design researchers [18–24]. The introduction of freeform surfaces can significantly improve the performance of off-axis reflective optical systems.

Obtaining a good starting point is an important step in the optical design process. Three methods can be used to obtain a starting point for a freeform off-axis reflective system. The first method is to search within lens databases and the existing literature. However, in the patents and literature involving off-axis reflective systems, most of the off-axis reflective systems are off-axis three-mirror systems or off-axis two-mirror systems. Off-axis multi-mirror systems with four or more mirrors are rare. Therefore, for off-axis multi-mirror systems with a relatively high number of mirrors, a suitable starting point may well not be found. The second method involves the creation of a coaxial spherical or aspherical system using paraxial optical theory. For coaxial reflective systems with different numbers of reflective surfaces, the designer needs to establish different aberration functions, which is a complex process. As the difference between the starting-point design obtained using this method and the optimum design is generally considerable, the optimization of such a starting point is generally a tedious and time-consuming process, which may fail to produce a satisfactory design result. The third method is to use direct design methods. Work has been conducted on design methods for freeform systems, which has made important contributions to the field of optical design, e.g., the partial differential equation method [25,26], the simultaneous multiple surface method [27] and the construction-and-iteration method [28,29]. The construction-and-iteration method is suitable for the design

of off-axis reflective systems that work with a specific object size and entrance pupil size. However, as Fermat's principle is used in the process of calculating the data points, it is very difficult—even impossible, sometimes—to design an optical system with more mirrors using the existing construction-and-iteration method. The reasons are as follows. First, in this method, when calculating the shape of a reflective surface, it is necessary to determine the minimum value of a complex non-linear optical path function, which is related to the shapes and number of the reflective surfaces that exist between the image plane and the surface being calculated. After the number of reflective surfaces of the system and the type of each reflective surface are given, it is a hard and time-consuming task to list the expressions of the optical path function corresponding with each reflective surface. Second, for a system with a relatively larger number of mirrors, it takes a longer time to find the minimum value of the optical path function through a search process. What is worse is that for a system with a high number of mirrors, the minimum value of the function often cannot be found through the search process, which causes the design process to be interrupted. Therefore, there is a lack of a simple and general design method for freeform off-axis reflective systems with relatively larger numbers of mirrors.

A freeform reflective system with more reflective surfaces offers greater design freedom and can realize systems with a higher performance. In this paper, we propose a general method to determine the appropriate starting point for a freeform off-axis multi-mirror system design that can promote the realization of high-performance reflective systems. Unlike the existing design methods, the method does not need to establish aberration functions or optical path functions nor does it need to solve the minimum value of aberration functions or optical path functions, which is highly suitable for the design of systems with a relatively high number of reflective surfaces. During the design process, the proposed data point calculation method was used to calculate the shapes of reflective surfaces, which were based on solving the ideal emergent rays from, and incident rays on, the reflective surface in sequence. Unlike the existing construction-and-iteration method, Fermat's principle was not used when calculating the data points used in the method proposed in this paper; this method can be applied to the design of multi-mirror systems with any number of mirrors. To ensure that the design method could cover the design of high-performance systems and moderate systems, a point-by-point design method that first gradually expanded the field of view (FOV) of the system and then gradually reduced the focal length of the system was proposed. Using the method, a starting point with specifications and an optical path structure that were similar to those of the desired design result were effectively obtained. Generally, when optimizing a design from such a starting point, the time required for the optimization process is shorter, the requirements placed on the design skills of the designers are reduced and the possibility of obtaining a satisfying optimization result is increased.

A general design method of freeform off-axis multi-mirror systems should cover the design of both high-performance freeform systems and moderate freeform systems as well as covering the design of freeform reflective systems with a relatively high number of mirrors and freeform reflective systems with a relatively low number of mirrors. Generally, the design of systems with a higher performance is more challenging and the design of reflective systems with a higher number of mirrors is more demanding. The design of an off-axis multi-mirror system with a relatively high number of mirrors and high performance was taken as an example to show the effectiveness of the proposed method. In the design example using the method proposed in this work, a starting point was obtained for the design of a freeform off-axis five-mirror system with a low F-number, for which the FOV was $10° \times 8°$, the entrance pupil diameter was 48.57 mm and the F-number was 0.7. After the optimization, an off-axis multi-mirror system design with a low F-number and good imaging quality was successfully obtained.

## 2. Methods

During the design process of a freeform off-axis multi-mirror system, calculating the data points on the surfaces of the system is an important step and is introduced in

Section 2.1. The point-by-point design method that first gradually expanded the field of view (FOV) of the system and then gradually reduced the focal length of the system as well as the specific steps of the design process are introduced in Section 2.2.

*2.1. Method for the Calculation of the Data Points*

This section outlines our proposed simple and general method to calculate the data points on the reflective surfaces of a freeform off-axis multi-mirror system, which is highly suitable for the design of systems with a relatively high number of mirrors.

Starting with a system in which each reflective surface was a plane, a freeform system was then obtained via a construction process. Taking this freeform system as the starting point, the imaging quality of the system could then be improved through an iteration process. The method of calculating the data points used in the construction process and the iteration process are introduced next, respectively.

The number of reflective surfaces of the system was denoted by $S$. According to the order in which they intersected with the incident rays, the reflective surfaces were called $\Omega_1, \Omega_2, \ldots, \Omega_S$ in sequence. During the construction process, the shape of each reflective surface in the system was calculated in order from the back to the front. The specific process was as follows.

1. Feature fields were selected and a specified number of feature rays from different pupil coordinates were selected from the rays in each feature field. According to the object–image relationship, the position of the ideal image point was calculated for each feature field. Using the nearest-ray algorithm, the positions and the normal vectors of the data points on the final reflective surface $\Omega_S$ in the system were then calculated. These data points were subsequently fitted to a freeform surface [28,30]. A system was then obtained in which the last reflective surface was a freeform surface; the remaining reflective surfaces were all plane surfaces.

2. The ideal emergent rays and the ideal incident rays were then calculated for the last reflective surface $\Omega_S$. The intersections of all the feature rays with surface $\Omega_S$ were solved and these intersections were then taken as the data points on surface $\Omega_S$. Point $I_t$ was the ideal image point for the $t$-th feature field. Point $D_m^{(t,S)}$ was the data point on the $S$-th reflective surface that corresponded with the $m$-th feature ray of the $t$-th feature field. The vector $n_m^{(t,S)}$ was the normal vector of surface $\Omega_S$ at point $D_m^{(t,S)}$. Ray $R_m^{(t,S)}$, which intersected with surface $\Omega_S$ at point $D_m^{(t,S)}$ and intersected with the image plane at the ideal image point $I_t$, was the ideal emergent ray that corresponded with data point $D_m^{(t,S)}$. Based on the directions of the ideal emergent ray $R_m^{(t,S)}$ and the normal vector $n_m^{(t,S)}$, the direction of the ideal incident ray $R_m^{(t,S-1)}$ that corresponded with point $D_m^{(t,S)}$ was then calculated. Using the same method, the ideal emergent rays and the ideal incident rays corresponding with all data points on surface $\Omega_S$ were then calculated.

3. The first data point on surface $\Omega_{S-1}$ was calculated. The ideal incident rays on surface $\Omega_S$ were the ideal emergent rays from surface $\Omega_{S-1}$. As illustrated in Figure 1, a ray $R_{m'}^{(t',S-1)}$ was selected from the ideal emergent rays from surface $\Omega_{S-1}$ and the intersection of ray $R_{m'}^{(t',S-1)}$ and surface $\Omega_{S-1}$ was taken to be the first data point $D_{m'}^{(t',S-1)}$ on surface $\Omega_{S-1}$. The actual direction of incidence $r_{m'}^{(t',S-2)}$ of a ray from the $t$-th feature field at point $D_{m'}^{(t',S-1)}$ was then obtained by ray tracing. The ideal normal vector $N_{m'}^{(t',S-1)}$ of $\Omega_{S-1}$ at data point $D_{m'}^{(t',S-1)}$ was calculated using the law of reflection. Data point $D_{m'}^{(t',S-1)}$ was then called the calculated data point. The plane that passed through point $D_{m'}^{(t',S-1)}$ and was oriented perpendicular to vector $N_{m'}^{(t',S-1)'}$ was called the tangent plane, which corresponded with data point $D_{m'}^{(t',S-1)}$.

4. The order in which the data points were to be solved was then determined. Each data point on surface $\Omega_{S-1}$ was located on an ideal emergent ray from surface $\Omega_{S-1}$ and had a one-to-one correspondence with the ideal emergent ray. Based on the distances between the ideal emergent rays from surface $\Omega_{S-1}$ and the first data point $D_{m'}^{(t',S-1)}$,

the order in which the data points corresponding with these ideal emergent rays should be calculated was then determined.

5. The position of a data point on surface $\Omega_{S-1}$ could then be calculated. Suppose, for example, that this data point corresponded with the ideal emergent ray $R_{m''}^{(t'',S-1)}$. To ensure the smoothness of the data points, the data point that corresponded with ray $R_{m''}^{(t'',S-1)}$ should be located on the tangent plane of a specific calculated data point that lay closest to ray $R_{m''}^{(t'',S-1)}$ [28]. The intersection of this tangent plane and ray $R_{m''}^{(t'',S-1)}$ was taken as the data point corresponding with ray $R_{m''}^{(t'',S-1)}$. The same method used in step 3 was then used to calculate the normal vector that corresponded with this data point. This step was repeated until all data points on surface $\Omega_{S-1}$ were resolved. These data points were then fitted to a freeform surface. In this way, using steps 2 to 5, the calculation of the shape of surface $\Omega_{S-1}$ was completed.

6. The data points on surface $\Omega_{S-2}$ were then calculated. The intersections of surface $\Omega_{S-1}$ and the ideal emergent rays from surface $\Omega_{S-1}$ were selected as the new data points on surface $\Omega_{S-1}$. Using the actual normal vector of $\Omega_{S-1}$ at these data points, the directions of the ideal incident rays corresponding with the ideal emergent rays from surface $\Omega_{S-1}$ were then calculated. These ideal incident rays on surface $\Omega_{S-1}$ were the ideal emergent rays from surface $\Omega_{S-2}$. The same method was then used to calculate the positions and normal vectors of the data points on surface $\Omega_{S-2}$; these data points were then fitted to a freeform surface. The shapes of the remaining reflective surfaces were then calculated using the same method.

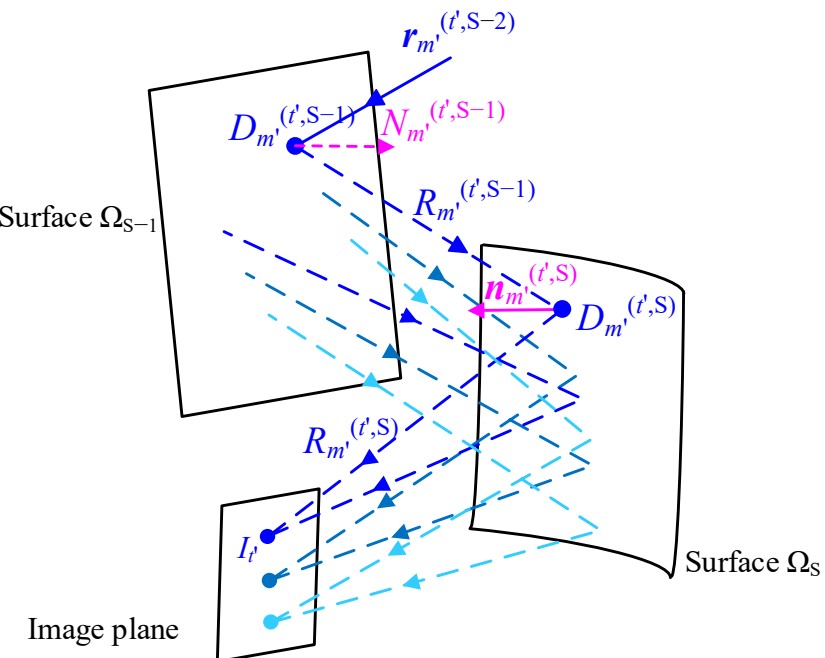

**Figure 1.** Calculation of data points during the construction process.

After all the surfaces of the system were constructed as freeform surfaces, the iteration process was used to improve the imaging quality of the system. The calculation method for the data points used in the iteration process is now introduced with the *p*-th reflective surface $\Omega_p$ of the system being taken as an example here. Using the same method that was used in the construction process, the ideal emergent ray $R_m^{(t,p)}$ from surface $\Omega_p$ was calculated. As shown in Figure 2, the position of the intersection of ray $R_m^{(t,p)}$ and surface $\Omega_p$ was selected as the position of data point $D_m^{(t,p)}$ on surface $\Omega_p$. The actual direction of incidence $r_m^{(t,p-1)}$ of a ray from the *t*-th feature field at point $D_m^{(t,p)}$ was obtained by ray tracing. The ideal normal vector $N_m^{(t,p)}$ of surface $\Omega_p$ at point $D_m^{(t,p)}$ was then calculated using the law of reflection. After all the data point calculations were completed, the shape

of surface $\Omega_p$ was then recalculated using the method that considered both the coordinates and the normal vectors of the data points [30]. Using this method, the shape of each reflective surface was recalculated in turn to improve the imaging quality of the system.

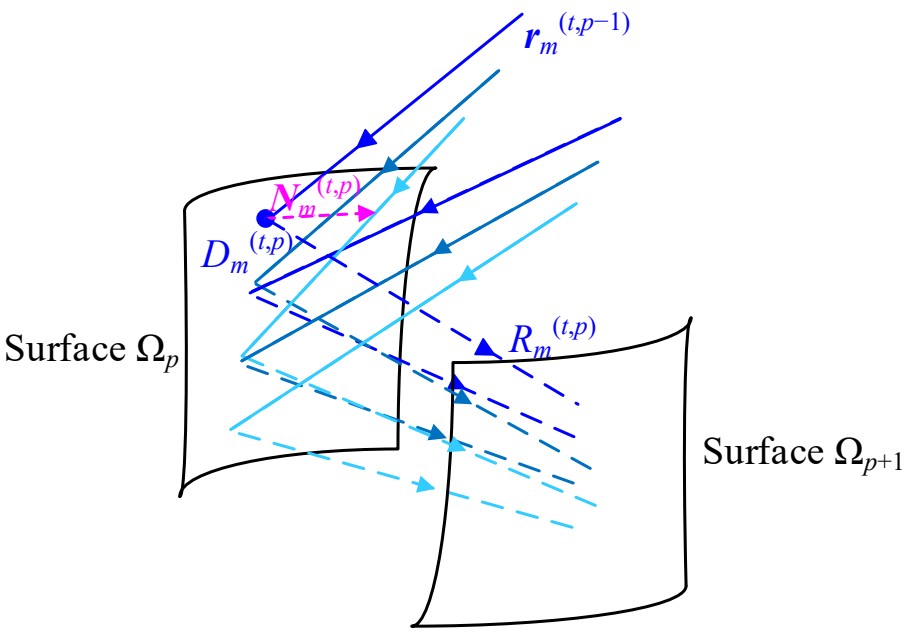

**Figure 2.** Calculation process for the data points during the iterative process.

### 2.2. Establishment of the Off-Axis Multi-Mirror System

This section describes how a good starting point for a freeform off-axis multi-mirror system can be obtained using the data point calculation method proposed in Section 2.1 and the proposed point-by-point design method that first expanded the FOV and then gradually reduced the focal length of the system.

The design of systems with a higher performance is more challenging. If a method can be applied to the design of moderate freeform reflective systems, the method may not necessarily be also suitable for the design of high-performance freeform reflective systems. To ensure that the design method could cover the design of high-performance systems, a point-by-point design method that first expanded the FOV and then gradually reduced the focal length of the system was proposed.

The design process of a multi-mirror system can be divided into the following three steps. The first step is to establish a planar system. The second step is to establish a multi-mirror system with an entrance pupil diameter and an FOV that meet the design requirements as well as a focal length that is greater than the design requirement. The third step is to gradually reduce the focal length of the system to obtain the required multi-mirror system. The entrance pupil diameter of the system remains unchanged throughout the design process.

In the off-axis multi-mirror system specifications used in this work, the FOV was set at $\varphi_x \times \varphi_y$ and the focal length was $f_0$. First, a strategy of the gradual expansion of the FOV [31,32] and the data calculation method proposed in Section 2.1 were used to establish an off-axis multi-mirror system in which the entrance pupil and the FOV both met the design requirements. The focal length was $\beta^K f_0$, where $K$ was a positive integer and $\beta$ was greater than 1. In this paper, the parameters $\beta$ and $K$ are called the focal length scaling parameters. A strategy of the gradual reduction in the focal length and the data calculation method proposed in Section 2.1 were then used to obtain an off-axis multi-mirror system with the required FOV of $\varphi_x \times \varphi_y$ and the focal length $f_0$. These two processes are introduced here in turn.

In the process of the expansion of the FOV, both the focal length of the system and the bias of the central field in the *y*-direction remained unchanged; the system also remained

symmetrical about the meridian plane. First, a planar system was established in which the entrance pupil and the FOV both met the system design requirements. The FOV of the planar system was then reduced to $(\varphi_x/Q) \times (\varphi_y/Q)$, where the positive integer $Q$ was the number of times that the FOV was extended. Using the method to calculate the data points proposed in Section 2.1, each reflective surface in the planar system was constructed as a freeform surface in turn. The FOV of the system was then uniformly expanded. During this process, the method of calculating the data points in the iteration process that was proposed in Section 2.1 was used. In the $q$-th iteration, the FOV of the system was given by $(q\varphi_x/Q) \times (q\varphi_y/Q)$. After the system iteration was performed $Q$ times, a freeform system was obtained in which the entrance pupil diameter and the FOV both met the design requirements. Its focal length was $\beta^K f_0$.

Beginning with the system that was obtained by expanding the FOV, the F-number of the system was then reduced further using the focal length reduction method proposed in this section; an off-axis multi-mirror system was obtained with the required focal length of $f_0$. During this process, the entrance pupil diameter and the FOV of the system both remained unchanged and the data point calculation method used in the iteration process in Section 2.1 was used. After the recalculation of the shapes of each of the reflective surfaces was completed, the focal length of the system was reduced to $1/\beta$ of its original value. The reflective surfaces in the system were named using the same naming conventions that were used in Section 2.1.

1. The focal length of the system was reduced to $1/\beta$ of its original value and the positions of the ideal image points for each feature field were calculated. The shape of the reflective surface $\Omega_S$ of the system was then calculated.

2. The system focal length was reduced again to $1/\beta$ of the focal length that was obtained in step 1 and the positions of the ideal image points for each feature field were calculated. Using the method proposed in Section 2.1, the ideal emergent rays from, and ideal incident rays on, the reflective surface $\Omega_S$ were then calculated. The ideal incident rays on $\Omega_S$ were ideal emergent rays from surface $\Omega_{S-1}$. The intersections of the reflective surface $\Omega_{S-1}$ and the ideal emergent rays from surface $\Omega_{S-1}$ were then selected as the data points on the reflective surface $\Omega_{S-1}$. Using the actual directions of incidence and the ideal emergent directions of the feature rays at these data points, the shape of the reflective surface $\Omega_{S-1}$ was then calculated.

3. The focal length of the system was reduced again and the positions of the ideal image points for each feature field were calculated again. The ideal emergent rays from, and ideal incident rays on, the reflective surface $\Omega_S$ and the reflective surface $\Omega_{S-1}$ were calculated in turn. The intersections of the reflective surface $\Omega_{S-2}$ and the ideal incident rays on $\Omega_{S-1}$ were selected as the data points on reflective surface $\Omega_{S-2}$; the shape of the reflective surface $\Omega_{S-2}$ was then calculated. Using the same method, the shapes of the remaining reflective surfaces were then calculated in turn.

4. Steps 1–3 were repeated until the focal length of the system was equal to $f_0$.

Using the method proposed, a starting point for the design of a freeform off-axis multi-mirror system could be automatically obtained after inputting a planar system, the specifications, the number of times the FOV was extended ($Q$) and the focal length scaling parameters ($\beta$ and $K$). The flowchart of the design process is shown in Figure 3, where $f$ denotes the focal length of the system and S is the number of reflective surfaces of the system.

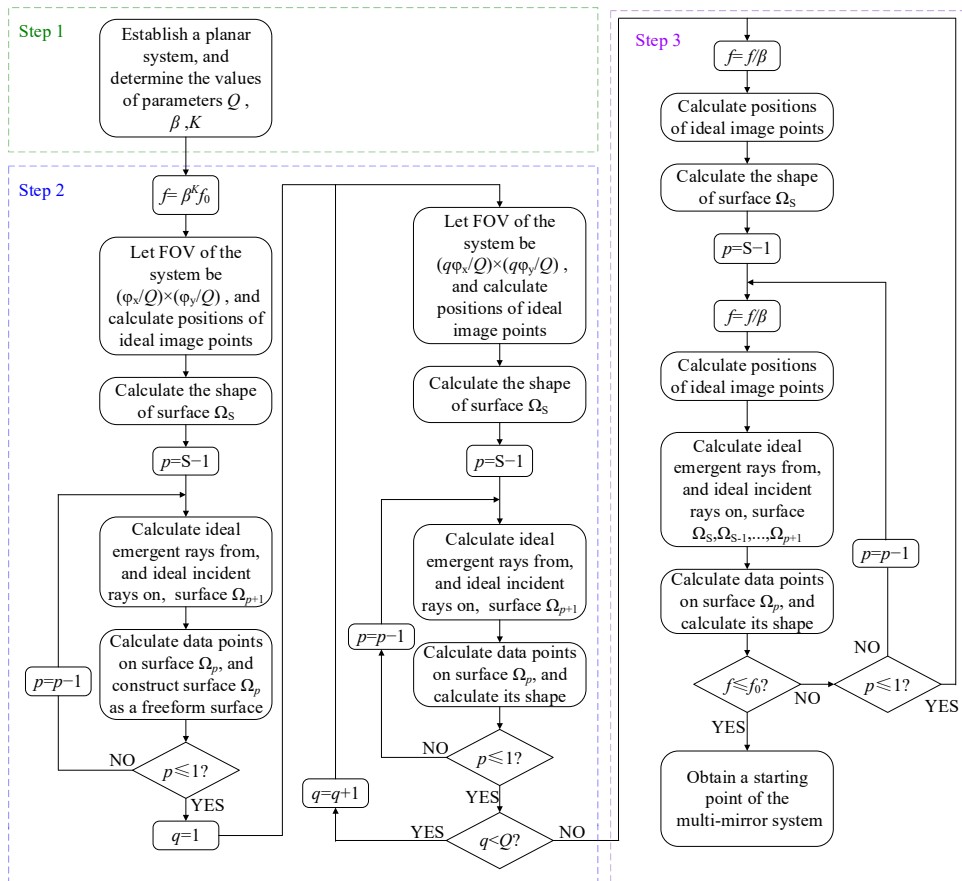

**Figure 3.** Flowchart of the design process.

## 3. Design Example

This section shows how to obtain a starting point for a freeform off-axis multi-mirror system using the method proposed. The design of high-performance freeform systems is more challenging than the design of moderate freeform systems and the design of reflective systems with a relatively high number of mirrors is more challenging than the design of reflective systems with a relatively low numbers of mirrors. To show the effectiveness of the proposed method, the design of a freeform off-axis multi-mirror system with a high performance and a relatively high number of mirrors was taken as an example, which was a freeform off-axis multi-mirror system with a low F-number. The system specifications are given in Table 1. Through several attempts, it was found that it was difficult for the three-mirror system and the four-mirror system to achieve such a high performance whereas the five-mirror system could achieve such a high performance. Therefore, an off-axis five-mirror system was designed, as detailed in this section. After a planar off-axis five-mirror system and the required parameters were input, a starting point for the design of an off-axis five-mirror system with a low F-number was automatically obtained using the method proposed in this paper. The type of freeform surface was an XY polynomial and its order was six.

**Table 1.** System specifications.

| Parameter | Specification |
|---|---|
| FOV | $10° \times 8°$ |
| F-number | 0.7 |
| Focal length | 34 mm |
| Wavelength | 8~12 μm |

First, an off-axis five-mirror system was established in which each reflective surface was a plane. It was required that the optical structure of the planar system was similar to that of the desired system and the planar system had no obstruction. The optical path diagram of this established planar five-mirror system is shown in Figure 4. Using the proposed method, a freeform system denoted by $M_1$ was then obtained in which the FOV was $10° \times 8°$, the focal length was 47.5887 mm and the reflective surfaces were all freeform surfaces. During the process of the expansion of the FOV, the FOV of the system expanded by $0.25°$ in the $x$-direction and $0.2°$ in the $y$-direction each time. The values of the focal length scaling parameters $\beta$ and $K$ were $1/0.9986$ and 240, respectively. The optical path diagram, the spot diagram and the distortion grid of system $M_1$ are shown in Figure 5. In the spot diagram, the unit of the RMS is millimeters. The maximum distortion in the $x$-direction was 1.9% and the maximum distortion in the $y$-direction was 6.1%.

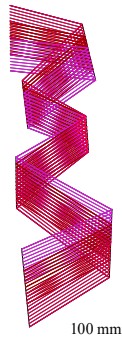

**Figure 4.** Planar system $P_0$.

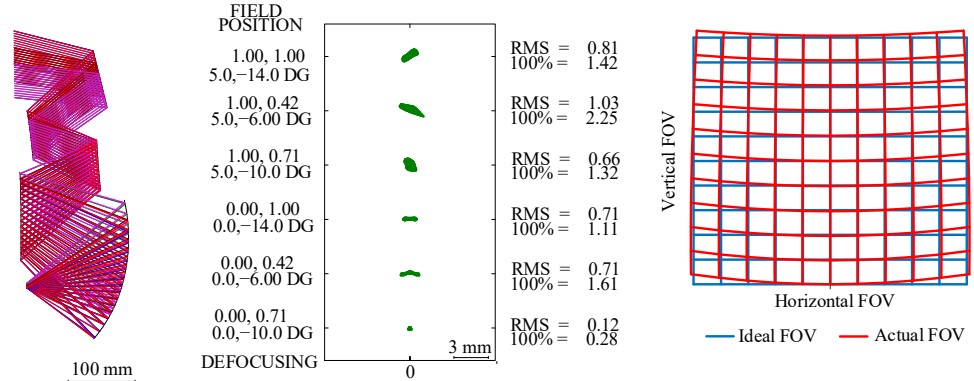

**Figure 5.** System $M_1$ obtained by extending the FOV of the system.

Starting from system $M_1$, the focal length of the system was then gradually reduced using the method proposed in Section 2.2. After the shapes of the reflective surfaces were recalculated 240 times, the focal length of the system was reduced from 47.5887 mm to 34 mm and the freeform system $M_2$ was obtained. The optical path diagram, the spot diagram and the distortion grid of system $M_2$ are shown in Figure 6. The maximum distortion in the $x$-direction was 9.2% and the maximum distortion in the $y$-direction was 9.7%. The system was a good starting point for the optimization process. The optical path structure of this system was similar to that of the final design result and the feature rays were basically concentrated at the ideal image points. During the design process from a planar system to a system with a low F-number, the optical power distribution and surface shapes of the system significantly changed, which resulted in system $M_2$ having an obstruction. The obstruction of the system could be easily eliminated by the subsequent optimization process. The designer could also choose to obtain a new unobscured freeform system by adjusting the positions of the mirrors of the initial planar system and repeating the design process.

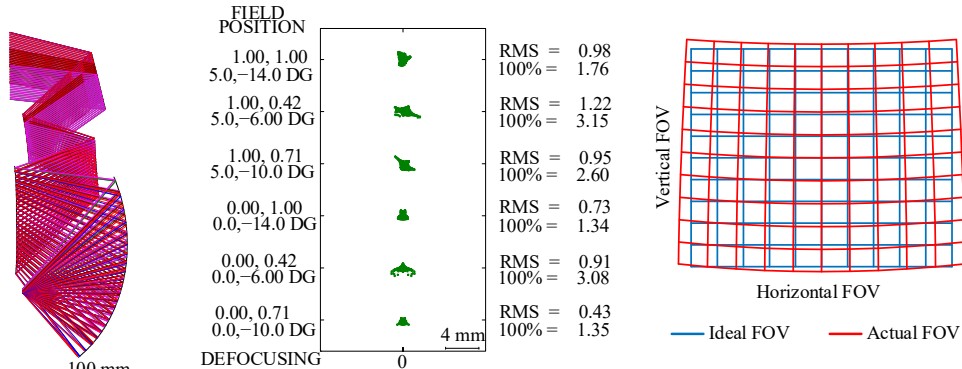

**Figure 6.** System $M_2$ obtained using the focal length reduction method.

System $M_2$ was then optimized. The optical path diagram, the root mean square (RMS) wavefront error and the distortion grid of the system after the optimization process are shown in Figure 7. The maximum value of the RMS wavefront error was 0.0821 $\lambda$ (where $\lambda$ = 10 μm) and the maximum absolute distortion was 4.7%.

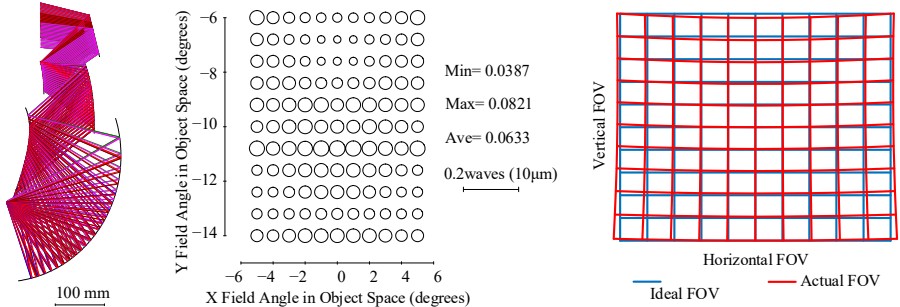

**Figure 7.** System design after optimization.

## 4. Discussion

To effectively obtain a good starting point for an off-axis multi-mirror system with a high performance, a method of first expanding the field of view and then reducing the focal length was adopted during the design process to gradually reduce the gap between the current system and the desired system. Other methods of gradually improving the performance of the system—e.g., the method of reducing the focal length first and then expanding the field of view—could also gradually narrow the gap between the current system and the desired design result and theoretically could be applied to the design of high-performance systems. After designing multi-mirror systems with low F-numbers using different methods to gradually improve the performance of the system, the method of first expanding the field of view and then reducing the focal length could generally obtain a better design result. Therefore, when designing a system with low F-number, the method of first expanding the field of view and then reducing the focal length is recommended. For other types of high-performance systems, such as systems with a wide field of view, other methods to gradually improve the performance of the system may be better; these may be studied in the future.

How we chose the values of parameters Q, β and K is now discussed. If the FOV of the system was relatively wide, choosing a relatively larger value for Q was suggested. If the F-number of the system was relatively low, choosing a relatively large value for K and a value that was relatively close to 1 for β was suggested. If the starting point obtained was unacceptable, the designer could select another set of values for those input parameters and repeat the design process. As the design process of the starting point required almost no participation of the designer, determining the values of these parameters according to design experience was feasible, which did not require much time of the designer.

The method proposed in this paper was a general design method for off-axis multi-mirror systems that could also be used to design reflective systems with other numbers of mirrors, such as off-axis six-mirror optical systems and off-axis four-mirror optical systems. For the design of the starting points of systems with other numbers of mirrors, only the number of mirrors of the initial planar systems needs to be changed. The optical path structure of the starting point obtained by this method was similar to that of the initial planar system. If an off-axis multi-mirror system with another optical path structure is expected to be obtained, it is only necessary to change the optical path structure of the planar system.

Compared with traditional design methods, the proposed method is a simple and general design method for multi-mirror systems that could obtain a good starting point for off-axis reflective systems with more mirrors and a high performance. A good starting point for an off-axis multi-mirror system could be automatically obtained after inputting an initial planar system and the necessary parameters. The FOV and entrance pupil diameter of the starting point were consistent with the design requirements and the optical path structure of the starting point was similar to that of the desired design result. As the gap between the starting point and the final design result was relatively small, optimization was generally easier and quicker.

## 5. Conclusions

The aim of this paper was to propose a simple and general design method of freeform off-axis multi-mirror systems that could promote the realization of high-performance reflective systems containing freeform surfaces. By taking a planar system as the starting point and then inputting the required parameters, a starting point for the design of a freeform off-axis multi-mirror system could be automatically obtained using the proposed method. An increase in the number of mirrors and improvements to the system performance will increase the design difficulty of the system. The design of a freeform reflective system with five mirrors and a high performance was taken as an example to show the effectiveness of the method proposed. The proposed method was a general design method of reflective systems and could also be used for the design of moderate reflective systems and reflective systems with other numbers of mirrors. The proposed method for the calculation of the data points could also be used in the design process of coaxial refractive systems, thus providing a tool for the design of coaxial refractive systems.

**Author Contributions:** Conceptualization, J.Z.; software, X.L.; writing—original draft, X.L.; writing—review and editing, J.Z. All authors have read and agreed to the published version of the manuscript.

**Funding:** This research was funded by the National Natural Science Foundation of China (NSFC) grant number 62175123.

**Institutional Review Board Statement:** Not applicable.

**Informed Consent Statement:** Not applicable.

**Data Availability Statement:** Not applicable.

**Conflicts of Interest:** The authors declare no conflict of interest.

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
