# Peer review of "Design Method of Freeform Off-Axis Multi-Mirror Optical Systems"

_photonics, doi:10.3390/photonics9080534_

Round 1

Reviewer 1 Report

In this paper, the CI direct construction method of optical system proposed by Zhu Jun's research group is adopted, and this method is extended to the design of multi mirror optical system. For the off-axis multi mirror optical system, there have been no papers on the direct design of multi mirrors before, which is innovative to some extent. In this paper, the author constructs the initial structure by CI method, and completes the design of off-axis five mirrors by gradually expanding the FOV and reducing the focal length. The method is described in detail, the simulation data is sufficient, and the design results are good, which can be used for reference by peers.

Some problems I concerned in the article as follows:

1. The author chooses the off-axis five-mirror system as the demonstration of the design method of the multi mirror system. What is the basis for the selection of the number of mirrors? Why is it not a four-mirror system, a six-mirror system or other multi-mirror systems? What is the essential difference between this design method and the classical three-mirror system?

2. Abbreviations of terms. For the first time in the text, give the abbreviation (capitalized English letter combination), which can be directly used in the following text, for example, "FOV" in lines 68 and 94

3. About the flow chart. It is suggested that when the author draws the flow chart, the contents in the chart correspond to the design steps described one by one. It is suggested to modify Figure 3

4. As for the expansion of the field of view and the reduction of the focal length, are these two steps in sequence or can they be carried out together?

5.In Fig.7 the final designed system, the image plane location is so close to the fourth mirror, and in the engineering application, how to consider to set the image plane devices?

Reviewer 2 Report

The authors proposed a design method for finding the starting point for multi-mirror freeform optical systems. This is achieved by setting up an initial planar reflective system, followed by gradually expanding FOV and shortening the focal length. The paper is publishable if the following concerns can be addressed:

  1. Please discuss the rationale for choosing the 5 mirror reflective system. Surely it is more difficult to design, but can the system specifications be satisfied using a simpler 3 or 4 mirror system? If so, why not use other techniques mentioned in the intro?

  2. System M2 seems to have obstructions of rays. Please explain.

  3. Why is the optimization procedure first expand the FOV then reduce the focal length? Can these steps be reversed or even interleaved? Is there any physical intuition in determining the order of this optimization procedure, or is it just empirically work better?

  4. How should one choose Q and beta? How are they related to the design specifications?

  5. The authors use the words “relatively high/low number of mirrors” in a number of places throughout the manuscript. This should be clearly defined: can >= 5 mirrors be considered as “relatively high”?

  6. Fig.5,6, what’s the unit of the RMS?

Round 2

Reviewer 2 Report

The authors have addressed all my concerns in the revised manuscript, and I recommend it to be published in its current form.

Author Response

We thank you sincerely for carefully reviewing the manuscript. Your comments are very helpful for improving the quality of the manuscript.